# Land Use Transition and Eco-Environmental Effects in Karst Mountain Area Based on Production-Living-Ecological Space: A Case Study of Longlin Multinational Autonomous County, Southwest China

**DOI:** 10.3390/ijerph19137587

**Published:** 2022-06-21

**Authors:** Min Wang, Kongtao Qin, Yanhong Jia, Xiaohan Yuan, Shuqi Yang

**Affiliations:** 1College of Urban and Environment Science, Central China Normal University, Wuhan 430079, China; qinkongtao@mails.ccnu.edu.cn (K.Q.); yuanxiaohan@mails.ccnu.edu.cn (X.Y.); sixsqy@mails.ccnu.edu.cn (S.Y.); 2Key Laboratory for Geographical Process Analysis & Simulation Hubei Province, Central China Normal University, Wuhan 430079, China; 3College of Environment and Resources, Guangxi Normal University, Guilin 541000, China; jiayh@mailbox.gxnu.edu.cn

**Keywords:** production-living-ecological space, karst mountain area, rocky desertification, land use transition, eco-environmental effects

## Abstract

The linkage mechanisms and optimization strategies between land use transition and eco-environmental effects that occur in the production-living-ecological space of karst mountain areas remain under-explored in the current literature. Based on county data collected in Longlin Multinational Autonomous County of Guangxi, which is located in the rocky desertification area of Yunnan, Guangxi, and Guizhou, this study contributes a county-level analysis on land use transition and eco-environmental effects by addressing two research questions: (1) Which factors of land use transition are related to the eco-environmental effects of production-living-ecological space? (2) What are the key land allocation mechanisms behind the interventions of local rocky desertification regulation policies? We conducted two sets of analyses to answer these two questions: quantitative analyses of the spatial and temporal evolution between land use transition, rocky desertification, and its eco-environmental effects, and qualitative analyses of policy interventions on production-living-ecological land development and rocky desertification management. The findings show that the occurrence of rocky desertification accompanied by unreasonable land use structure transition and its important factor is caused by ecological land being restricted by production-living land. Specifically, urbanization strategies coordinating ecological and socio-economic effects is significant to karst mountain areas. Moreover, the orderly increase of woodland slows down rocky desertification. Policies of “returning farmland to forest” and “afforestation of wasteland” have significantly reduced rocky desertification that can be applied to other geographical situations.

## 1. Introduction

In May 2019, the Opinions of establishing territorial spatial planning system and its implementation clearly stated that the main goal of territorial spatial planning is “to comprehensively improve the territorial space governance system and governance capacity by 2035, and to basically form a territorial spatial pattern with an intensive and efficient production-living-ecological space”. This means that the development model of land space in China has changed from production space to the coordinated development model of production-living-ecological space. In January 2022, the Ministry of Natural Resources and the National Standardization Administration Committee issued a “Three-Year Action Plan to construct Technical Standards System for Territorial Spatial Planning (2021–2023)”, which further clarifies that strengthening spatial coordination and coordinating production-living-ecological space are the basic principles for the overall protection and development of territorial space. It can be seen that the fundamental starting point of organizing land use in territorial spatial planning is to reduce the interaction between land use and the environment through the rational arrangement of various land types. Especially in the face of environmental degradation, resource transitional exploitation, and inadequate living space caused by the imbalance of territorial space development, this study helps to analyze the root causes of the problems. Further exploring the interactive relationship between land use transition and its eco-environmental effects has important theoretical and practical significance for promoting the construction of ecological civilization and optimizing the patterns of production-living-ecological space.

Land use transition reflects natural environment [1] changes and socioeconomic development [2]. Global warming, carbon cycle alteration [3], ecological degradation [4], biodiversity loss [5], ecosystem health [6], and the imbalance of the sea-land water cycle [7] caused by land use transition have been under concern by scholars. The existing studies mainly have two aspects: the process studies about land use characterization [8], change prediction [9], and drivers [10], and the mechanism studies on land use national forest transition [11], regional arable land transition, and urban-rural construction land transition [12,13,14], which effects urban-rural development [15], resources [16], and eco-environmental impacts [17]. Specifically, the land use transition under the background of production-living-ecological space is manifested in the land, leading to function change, which is focused on the transition among the three leading functions of production-living-ecological space [18,19]. In the research on the production-living-ecological space patterns of land use and its eco-environmental effects, investigators mainly used the methods of land use transition dynamics, transfer matrix, driving force index, center of gravity transfer model, eco-environmental quality index, ecological contribution rate. They took Yangtze River Delta [20], Northeast China [21], coastal areas [22], and Yellow River Basin [23] as case study areas to discuss the changes of regional eco-environmental quality. These research results showed eco-environmental impacts are mainly concentrated in two aspects. One is the impact on a single environmental factor (soil, carbon emissions, water quality, biology and climate, etc.) [24,25,26], and the second is the eco-environmental effects caused by land use on large and medium scales, including ecosystem value [27], ecological vulnerability [28], and eco-environment deterioration [29]. These studies provide a theoretical framework and research basis for analyzing the temporal change process of land use, and provide a useful reference for this study. However, it is very necessary to describe the linkage mechanism between the ecological process of land use transition and the functional transition of production-living-ecological space. In particular, the case study of karst mountain areas with man-land prominent contradictions in Southwest China should be further explored. Based on this, this study connects the evolution of production-living-ecological space with land use transition to explore the ecological effect of land use in karst mountain areas.

The eco-environment of karst mountain areas is relatively fragile. Rocky desertification has caused decreases in surface water, vegetation restoration difficulties, and geological hazards of droughts, floods, mudslides, and ground subsidence [30]. On a larger scale, it can even affect the carbon balance [31] and regional climate conditions [30]. Studies on the process and environmental benefits of deforestation and reforestation in Southeast Asian countries [32,33], South American mountain ranges [34], Zambia [35], and the Amazon [36] show that irrational land use and reckless deforestation forming soil erosion [37] are the most fundamental causes of rocky desertification. The unreasonable land use transition in this special area not only leads to the deterioration of rocky desertification, but also leads to the reduction of available land types. The weakening of production functions and ecosystem imbalance could cause continuous poverty in karst mountain areas [38,39]. Therefore, we analyzed the territorial spatial changes of Longlin Multinational Autonomous County from 2005 to 2020. The research aims were (1) to determine the direction of land use transition by analyzing land type changes through remote sensing images in order to clarify the action mechanism and competition process among the three functional spaces of production-living-ecological space and assess the evolutionary direction of rocky desertification and the quality of eco-environment, and (2) to accurately grasp the evolution and distribution characteristics of the three functional spaces and their driving mechanisms, which provide case support and countermeasure suggestions for territorial space optimizing, poverty alleviation, and high-quality development.

## 2. Theory Background

### 2.1. Land Use Transition Theory

Land use transition is the temporal change of national or regional land use form. It usually corresponds to the stage of economic and social development. Mather, a geographer at the University of Aberdeen in Scotland, first proposed the forest transition or forest area transition hypothesis [40]. He believes that most forestry countries have to go through a continuous stage of land use transition until a new balance is established. At the end of the transition, the woodland can be increased again by self-regeneration and artificial afforestation. Inspired by Mather, Grainger [41] proposed land use transition theory. At the beginning of the 21st century, it was introduced in China by Long Hualou [19]. Since then, land use transition has become a common method used to explore the evolution of the human-land relationship [42], and the main contents of this research are the driving forces of land use change [43], dynamic land use evolution process [20], and development prediction and environmental impact assessment [44]. In recent years, many scholars have used landscape diversity indices, fragmentation, fractal dimension, and other indicators to describe the state of land use and spatial change trends [45]. The land use comprehensive index model and land use transfer matrix are used to reflect the transition state of temporal and spatial change [20,21,22,23], and the social driving force model, IMPL integration model, AIM model, deforestation model, and economic theory model have been introduced to deeply analyze the causes, processes, and prediction of land use transition [46], which have become the main research methods of land use transition.

In addition, different land cover has a significant impact on the degree of rocky desertification [47]. Research shows that land use and human settlements are important factors determining rocky desertification [48,49]. Social and economic activities significantly change the forward and reverse evolution direction of rocky desertification. Unreasonable spatial activities, such as steep slope planting, overgrazing, and deforestation are the key factors affecting rocky desertification [50]. Ecological projects, such as returning farmland to forest, afforestation, and comprehensive control of rocky desertification, can effectively control and restore rocky desertification [51]. The karst mountain area in southwest China is located in the center of the karst area in eastern Asia. It is not only a large contiguous area, but also a traditional agricultural farming area. Compared with the karst mountain areas of central and southern Europe and eastern North America, the karst mountain areas in southwest China have a smaller population density and weaker supporting capacity of the eco-environment. Therefore, exploring the relationship between land use transition and the evolution of rocky desertification levels can provide scientific reference for territorial space development, regional eco-environment protection and sustainable development.

### 2.2. Theory of Sustainable Land Use

It is a global issue to explore ways of sustainable land use to achieve rational allocation of resources. In 1993, the Food and Agriculture Organization of the United Nations (FAO) issued the Outline of Sustainable Land Use Assessment. It pointed out that soil quality should be maintained without degradation in the ecological aspect, and that efficient and rational utilization allocation should be ensured in the economic and social aspects. Moreover, the principle of intergenerational equity should be followed for contemporary needs, which have become the current guiding principles for sustainable land use [52]. According to P.J. Fu et al. [53], sustainable land use can be evaluated in three aspects: ecological, economic, and social. On the basis of this idea, the three aspects are linked based on the connotation and objectives of sustainable land use. Based on qualitative and quantitative analysis of the stability, for the development and sustainability of production-living-ecological land use patterns within a certain period of time, research is the core part of applying the theory of sustainable land use to practice.

### 2.3. Environmental Effects of Land Use Transition 

In 1856, Marsh and Lowenthal mentioned the linkage between land use and ecology in their book “Man and Nature”, which laid the foundation for the study of the ecological effects of land use transition. Land use transition is usually a purposeful transition based on certain development goals, the transition process will inevitably affect eco-environmental changes, and the two interact with each other to produce eco-environmental effects. The positive ecological effect mainly aims at managing and improving the eco-environment, and adapting the dominant functional type of land use to achieve ecological sustainability, but the negative ecological effects mainly refer to the deterioration of the eco-environment caused by the over-exploitation of land resources due to human activities. Currently, research on land use transition and its eco-environment effects focuses on two aspects, including the influence of single elements, such as plant and animal communities, climate, hydrology, and soil [24,54], and combined effects, such as the evolutionary effects of ecosystem service functions [27] and the spatial and temporal divergence effects of landscape patterns [55]. In terms of territorial spatial planning, research on the ecological planning concepts related to ecological functions of land use and structural layout [56], the ecological infrastructure [57], and the urban-rural ecosystem restoration [58] have also been strengthened. The eco-environmental effects of land use transition are an important way to understand and predict the status and change characteristics of regional eco-environmental quality, and are the basis for the preparation of territorial spatial planning and land use structure optimization, with important theoretical value and practical significance. Based on the eco-environmental effect evaluation model used to explore the eco-environmental effect response mechanism under the land use transition of karst mountain areas in southwest China, this paper provides a reference for the optimal layout of territorial space.

## 3. Materials and Methods

### 3.1. Study Area

Longlin Multinational Autonomous County is located in the northwestern part of Baise City, Guangxi, southwest China, which belongs to the Yunnan-Guangxi-Guizhou rocky desertification area (Figure 1). The county has 6 towns and 10 townships and a total area of 3551 km^2^. About 65% of the study area has a land slope greater than 15 degrees, with about 21% of the area having a land slope greater than 25 degrees, and the altitude ranges from 380 to 1950 m. The topography of the study area is dominated by mountains, hills, and depressions, which makes it a typical mountainous county, and most of the area is a rocky desertification area. The largest area of carbonate rocks is exposed, mainly containing continuous limestone or a limestone and dolomite combination. Karst landforms are widely developed; the main types are peak cluster depressions, peak cluster mountains, and also some peak cluster valleys [59]. The county belongs to the subtropical monsoon climate zone, which is warm and humid with abundant rainfall and an annual average temperature of 18~22 °C. Annual precipitation ranges from 1113 to 1713 mm, and precipitation in the region is mostly concentrated from April to September, with strong erosion effects and frequent occurrence of geological disasters, such as mudslides and landslides. The region has a dry season from October to March (it is often threatened by drought) with complex hydrological conditions and a high number of underground rivers [60]. The eco-environment of the study area is very fragile due to its remote location, concentration of minority groups, complex terrain, thin soil layer, and serious rock desertification. At the same time, the problems that lead to poverty, such as difficulties in schooling, work, income generation, and inconvenient transportation occur frequently, make this area one of the four extremely poor counties in Guangxi. There are relatively few farmland areas and low production, features which are mainly distributed in the canyon area where people live. Therefore, there are conflicts between agricultural development and ecosystem protection in these areas [61].

### 3.2. Date Sources

This study used four periods of land use data for 2005, 2010, 2015, and 2020 from the Resource Data Center of the Chinese Academy of Sciences (http://www.resdc.cn/Default.aspx, accessed on 1 December 2021), and the spatial resolution of the data are 30 m. We classified land types into 6 major categories, arable land, woodland, grassland, construction land, water, and unused land, which can be divided into 25 subcategories [62]. In this paper, land use data for Longlin County were divided into 3 primary categories and 8 subcategories (Table 1).

We used DEM data. The terrain slope was calculated by ArcGIS software. The DEM data source is a geospatial data cloud (http://www.resdc.cn/Default.aspx, accessed on 1 December 2021) and the spatial resolution of the data are 30 m.

The bedrock exposure data and vegetation cover data are both from Landsat series remote sensing satellites. Based on ENVI 5.3, the image data were preprocessed with radiometric calibration, geometric correction, image fusion and cropping, and then the NDVI and NDRI were calculated using the band calculator. The vegetation cover and bedrock exposure were obtained by calculation. In this work, the vegetation cover rate, rock exposure rate, and topographic slope were used to classify the rocky desertification level [63,64]. The map algebra tool of ArcGIS10.2 software was used to overlay and calculate the three datasets and classify the rocky desertification level [65]. Then, natural intermittent points were used to classify the study area into areas without rocky desertification, potential rocky desertification areas, mild rocky desertification areas, moderate rocky desertification areas, and severe rocky desertification areas. The urbanization rate data used in the paper were obtained from the Guangxi Statistical Yearbook, and the urbanization rate of Longlin County from 2005 to 2020 was obtained using the ratio of urban population to resident population.

### 3.3. Methods

According to the three logical contents of land use transition, the spatial and temporal pattern evolution of rocky desertification and ecological effect, we analyzed the characteristics of land use transition, rocky desertification evolution, and ecological effect response for the case study area from 2005 to 2020, using quantitative methods, such as the dynamic attitude model, the transfer matrix model, the eco-environmental quality index, geostatistical analysis, and the ecological contribution ratio.

#### 3.3.1. Land Use Transition and Evolution of Rocky Desertification

In order to explore the influence of land use changes on the regional eco-environmental quality, the eco-environmental quality index factors for each type of land use were assigned, with reference to other scholars [18,21], and adjusted according to the proportion of secondary land use of each functional land use type in Longlin County, so as to obtain the eco-environmental quality index factors for production-living-ecological space land use types (Table 1). Based on this, the following research steps were carried out.

(1)Single land use and rocky desertification dynamic attitude

It is mainly used to represent the rate and magnitude of change of different land use types and the rocky desertification occupation within the study area during the study period. In this study, a single dynamic attitude was used to analyze the rate and magnitude of change of a single functional type of land or rocky desertification, and the mathematical model was expressed as follows.
(1)$$Q_{i}=\frac{K_{b}-K_{a}}{K_{a}}\times \frac{1}{T}\times \frac{1}{100}$$

*Q_i_* is the dynamics of land use type and rocky desertification in a selected study period in the study area; *K_a_* and *K_b_* are the areas occupied by a land type and rocky desertification at the beginning and end of the study period in the study area, respectively; and *T* is the duration of the study.

(2)Land use and rocky desertification transfer matrix

The land use and rocky desertification transfer matrix is a matrix arranged according to the transition between different land use types and the rocky desertification land area. Its mathematical expression is:(2)$$X_{ij}=\left[\begin{array}{cc} \begin{array}{cc} X_{11} & X_{12}\\ X_{21} & X_{22} \end{array} & \begin{array}{cc} \cdots & X_{1n}\\ \cdots & X_{2n} \end{array}\\ \begin{array}{cc} \vdots & \vdots \\ X_{n1} & X_{n2} \end{array} & \begin{array}{cc} & \vdots \\ \cdots & X_{nn} \end{array} \end{array}\right]$$

In Equation (2), *X_ij_* represents the area, *n* represents the total number of land types or rocky desertification types, and *i* and *j* represent the land types or rocky desertification types at the beginning of the study period and at the end of the study period, respectively.

#### 3.3.2. Regional Eco-Environmental Quality

(1)Environmental quality index of ecological units

Due to the high scale-dependent characteristics of the spatial distribution pattern, in order to obtain the most suitable scale for the study, reference to the research results of Qingke Yang [20], Qi Jia [66], Yu Zhou [67], etc., the study area was debugged several times, and finally a 500 m × 500 m square was used to sample the study area with equal spacing, and 14,521 sample areas were generated. The environmental quality indices of the unitary sample areas were calculated as follows:(3)$$EV_{k}=\overset{N}{\underset{i=1}{\sum }}\frac{A_{ki}}{A_{i}}R_{i}$$

*EV_k_* is the ecological quality index factor of the *k*-th ecological unit, *R_i_* is the ecological quality index of the *i*-th land use type, *A_ki_* is the area of land use type i within the *k*-th ecological unit, *A_k_* is the area of the *k*-th ecological unit, and *N* is the number of land use types.

(2)Geostatistical analysis

The environmental quality indices of 14,521 ecological units were assigned to the center points of the sample area, and the quality values of the sample points were spatially interpolated using the semivariance function method to obtain the ecological quality distribution map of the whole study area. The statistical analysis equations are as follows:(4)$$\gamma \left(h\right)=\frac{1}{2n\left(h\right)}\sum_{i=1}^{n\left(h\right)}{\begin{array}{c} \left[Z\left(x_{i}+h\right)-Z\left(x_{i}\right)\right] \end{array}}^{2}$$

In Equation (4), *γ(h)* is the semivariance; *h* is the sample distance; *Z(xi)* and *Z(xi + h)* are the ecological quality values located at *xi* and *xi + h*, respectively; and *n(h)* is the total number of sample pairs with distance *h*. A semivariance function was used to fit the data, and based on this, a kriging method was used to spatially interpolate the eco-environmental quality index of Longlin Multinational Autonomous County and classify it into five levels, namely lowest quality areas, lower quality areas, general quality areas, higher quality areas, and highest quality areas.

(3)Ecological contribution of land use transition

The ecological contribution of land use change type refers to the change in regional ecological quality caused by a particular land use type change and is expressed as follows:(5)$$LEI=\left(LE_{t+1}-LE_{t}\right)LA/TA$$

In Equation (5), *LEI* is the ecological contribution of land use function transition; *LE_t_* and *LE_t+1_* are the ecological quality indices of a certain land use change type at the beginning and end of the change, respectively; *LA* is the area of the change type; and *TA* is the total area of the region.

## 4. Results

### 4.1. Land Use Transition Characteristics of Production-Living-Ecological Space in Longlin County

#### 4.1.1. Spatial Characteristics of Production-Living-Ecological Space Land Use 

From 2005 to 2020, the land use structure of Longlin County was dominated by ecological space, followed by production space with living space occupying the least area, and the average values of the total area ratio were 88.9%, 10.9%, and 0.19%, respectively. The ecological space was mainly composed of woodland and grassland, the production space was mainly agricultural space, the industrial and mining production space was gradually increasing from none to one, and the living space was mainly urban living space (Table 1). The distribution of production-living-ecological space is characterized by an interlocking pattern of ecological space surrounding production space and living space sporadically distributed in production space (Figure 2).

In terms of topographic slope effect, the altitude of Longlin County was distributed between 380 m and 1950 m, meaning it is mountainous. The ecological land in the water area was mainly distributed in the area below 800 m. Industrial and mining production land, urban living land, and rural living land were mainly distributed in the area of 400–800 m. Agricultural production land was mainly distributed in the area of 400–1200 m. Other ecological lands were distributed in the area of 800–1200 m. Grassland and woodland ecological lands did not vary significantly with elevation. Water ecological land was mainly distributed in the low slope range of 0–6°, which is in line with the law of water distribution in the low mountainous area in the south. Agricultural production land was mainly distributed in the 0–15° slope area. Industrial and mining production land and urban living land were distributed in the 0–6° slope area. The rural land for living was distributed in the 6–15° slope range. Ecological grassland and ecological woodland did not differ significantly in all slope areas. This indicates that these two land types are not significantly affected by slope. This indicates that urban construction, human activities, and production activities in the county are basically carried out in a low-elevation, low-slope, and low-hill area.

#### 4.1.2. Evolution of the Land Use Scale of Production-Living-Ecological Space 

According to equation 1, the rate and magnitude of single land type changes can be obtained. The results show (Table 2) that the urbanization process accelerated from 2005 to 2020, and Longlin County increased its regional infrastructure and income-generating industries. Urban and rural living land and industrial and mining production land continuously expanded to restrict other spaces. Among them, the areas of urban living land, rural living land, ecological land in water, and industrial and mining production land significantly changed and increased, with the most obvious changes in rural living land and industrial and mining production land. The area of grassland and ecological woodland, agricultural production land, and other ecological land areas all decreased.

In terms of ecological space, the overall decrease in ecological land area from 2005 to 2020 was 3.66 km^2^, with a change of −0.005%. Among them, the area of grassland continued to decrease, with a dynamic attitude of 0.315 from 2005–2010, to 0.011 from 2010–2015, and then to 0.133 from 2015–2020, and the disturbance of grassland continued to slow down. The overall decrease in woodland area shows an overall accelerated decrease, with a kinetic attitude of 0.063 from 2005–2010, to a small increase from 2010–2015, and then to 0.073 from 2015–2020, which indicates that woodland was being restricted by other land. The area of water land had a continuous slow increase with a small decrease from 2010–2015, but this shifted from 9.308 in the period from 2005–2010 to 5.371 from 2015–2020, which indicates that the county was effective in increasing the infrastructure of water ecology. The overall decrease of other ecological land indicates that the county made great efforts to reclaim unused land to meet the use of other land types of “productive, living and ecological space”.

In terms of production space, the productive land decrease from 2005–2020 was 2.2 km^2^, with a change of −0.04%. Agricultural production land continued to decrease, and its dynamic attitude changed from 0.098 in the period from 2005–2010, to 0.125 from 2010–2015, and then to 0.193 from 2015–2020, indicating that although agricultural production land had been decreasing, the rate of reduction was slow and the way of living was mainly based on agricultural production. Mining production land increased 5.8 km^2^ in 2020 and continued to increase. Its kinetic attitude increased to 29.33 in 2015, and its rate of increase continued to increase before 2015, with a kinetic attitude of 11.35 in the period from 2015–2020, and then the growth rate of mining production land area slowed down. Overall, the area of mining production land increased, the county’s industrialization continued to develop, and the government increased investment in industrial infrastructure construction.

In terms of living space, the living land increase was 4.8 km^2^, with a change of 17.78%. The area of urban living land increased from 1.7 km^2^ to 4.2 km^2^, and its dynamic attitude decreased from 12.94 in the period from 2005–2010 to 2.14 from 2010–2015, then increased to 7.097 in the period from 2015–2020, which indicates the urbanization process of the county continued to advance. The rural living land increased substantially from 0.1 km^2^ in 2005 to 2.4 km^2^ in 2020. In terms of dynamic attitude, the area of rural living land increased at the fastest rate from 2005–2010, but slowed down significantly from 2010–2020.

#### 4.1.3. Evolution of Production-Living-Ecological Space Land Use Transition Direction

According to Equation (2), the spatial analysis was combined with Arcgis version 10.2 (ESRI, Inc., Redlands, CA, USA.) to construct the land transfer matrix of production-living-ecological space for different time periods (Figure 3). 

From 2005 to 2020, the land use transition is characterized by the mutual transition of ecological land and production land, and the production and ecological land was transferred out to living land. Among the ecological land that was transferred out, woodland, grassland, and water areas were transformed into agricultural production land and a small part was transformed into industrial and mining production land and rural living land; water ecological land was mainly transformed from ecological grassland and urban living land was mainly transformed from ecological woodland. Among the productive land that was transferred out (mainly agricultural production land), most of it was transformed into ecological land, mainly ecological woodland, and a small portion into rural living land.

From 2005 to 2010, all other ecological land areas were transformed into water ecological land and agricultural production land. Some of the woodland and grassland ecological land was converted into industrial and mining production land, and the industrial and mining industry began to develop gradually. The county seriously responded to the policy of “returning farmland to forest” by converting woodland and agricultural production land in equal amounts. From 2010 to 2015, 0.1 km^2^ of woodland was transferred to other ecological land, which is the unused land formed by the bare surface of soil after logging occurs in the area. The area of agricultural production land transferred to woodland achieved a positive effect and the county’s implementation of “returning farmland to forest” began to bear fruit. The area of industrial and mining production land increased by 113.3% and the development strategy of “developing the county with industry” was implemented during the period to promote urbanization. From 2015–2020, all other ecological land was transformed into grassland and ecological woodland, which was the result of the restoration of bare soil land for planting. The area of ecological woodland and agricultural production land were almost equal to each other. Abandoned industrial and mining land was transformed into grassland and ecological woodland. Overall, the areas of urban living land, industrial and mining production land, and rural living land increased significantly during the period, and the county’s urbanization process accelerated, increasing the construction of living infrastructure services.

In conclusion, from 2005 to 2020, affected by the policies of “returning farmland to forest”, “afforestation of wasteland”, and “returning farmland to Lake”, the agricultural production land was reduced and transformed into woodland and ecological grassland, and the reforestation of unreasonable agricultural production land was implemented to build a stable ecosystem. Through reasonable territorial spatial planning, the mountain county took into account the coordinated effect of ecological restoration and economic development to promote the development of healthy urbanization.

### 4.2. Evolutionary Characteristics of Rocky Desertification in the Production-Living-Ecological Space of Longlin County

#### 4.2.1. Evolution of the Rocky Desertification Scale in Production-Living-Ecological Space 

Combining with equation 1, rocky desertification level share data and area change data (Table 3) from 2005–2020 were obtained. In the three time periods, the spatial level of rocky desertification in the study area was mainly dominated by potential rocky desertification, mild rocky desertification, and moderate rocky desertification. In 2005, the rocky desertification phenomenon was remarkable, and the area with the level of mild rocky desertification or above accounted for 52.8%. The ratio of rocky desertification area was mild rocky desertification > potential rocky desertification > moderate rocky desertification > severe rocky desertification. In 2010, the degree of rocky desertification was aggravated and the area of mild rocky desertification or above accounted for 72.8%. The area of severe rocky desertification increased significantly and the ratio of rocky desertification area was moderate rocky desertification > mild rocky desertification > potential rocky desertification > severe rocky desertification. From 2005 to 2010, the overall socioeconomic level was relatively backward. During this period, the county experienced three consecutive years of the once-per-century special drought, as well as poverty pressure that led to indiscriminate logging by local farmers, vegetation destruction, and bedrock exposure, causing a deterioration of rocky desertification in the study area. From 2010 to 2020, the rocky desertification in the study area showed continuous improvement, and the area of no rocky desertification and the potential rocky desertification area greatly increased, while the area of mild rocky desertification, moderate rocky desertification, and severe rocky desertification greatly decreased. In 2020, the study area was dominated by no rocky desertification area and its area share is 33.12%.

In summary, the areas without rocky desertification and with potential rocky desertification increased and the area with mild rocky desertification, moderate rocky desertification, and severe rocky desertification decreased from 2005 to 2020. In terms of the dynamic attitude of rocky desertification change, the speed of the increasing area of no rocky desertification and potential rocky desertification was accelerated, the speed of decreasing area of moderate rocky desertification was accelerated, and the speed of decreasing area of mild rocky desertification and severe rocky desertification had a decreasing trend. Longlin County is a typical karst mountain area, and its rocky desertification is more serious and involves a wide range of area. Factors such as the concentration of ethnic minorities, low production levels, and a shortage of resources have led to the destruction of vegetation and intensification of rocky desertification in the county, gradually forming a vicious cycle of rocky desertification—ecological degradation—population poverty.

#### 4.2.2. Rocky Desertification Level Area Transfer Matrix

In order to understand the conversion direction and scale of rocky desertification in different degrees, the spatial superposition operation was performed on the data of spatial and temporal distribution of rocky desertification in four periods according to equation 2, and a total of four periods of rocky desertification transfer matrix in Longlin County from 2005 to 2020 was obtained through statistical analysis (Figure 4). From 2005–2010, the direction of rocky desertification transfer was from a low level to a high level, and the degree of rocky desertification was intensified. A total area of 315.2 km^2^ of potential rocky desertification was transferred to mild rocky desertification, 281.5 km^2^ of mild rocky desertification was transferred to moderate rocky desertification, and 104.4 km^2^ was transferred to severe rocky desertification; 203.4 km^2^ of moderate rocky desertification was transferred to severe rocky desertification. From 2010–2020, the rocky desertification management in the study area was quite effective. Since 2011, the county has been actively implementing the comprehensive management of rocky desertification and the direction of rocky desertification transfer is generally high to low. In 2015, 336.1 km^2^ was transferred from mild rocky desertification to potential rocky desertification, and 261.4 km^2^ was transferred from moderate rocky desertification to mild rocky desertification. In 2020, 384.2 km^2^ was transferred from potential rocky desertification to no rocky desertification. The total area of rocky desertification in the study area decreased, the area without rocky desertification increased, and the rate of rocky desertification area reduction accelerated, indicating that the implementation of the rocky desertification comprehensive management project in Longlin County had greater effectiveness.

#### 4.2.3. Evolution of the Spatial Distribution of Rocky Desertification in the Production-Living-Ecological Space

The spatial distribution of rocky desertification in the study area showed a macro pattern that was serious in the south-central and northwestern parts and not serious in the southwestern corner and eastern part (Figure 2), and the rocky desertification landscape was fragmented. The topography of the study area was high in the south and low in the north, with many middle and high mountainous areas and many karst canyons developing in the south-central and northwestern parts, as well as unstable surface materials, serious precipitation erosion, frequent human activities, and a high rock exposure rate, leading to serious rock desertification. As shown in Table 3, rocky desertification areas in Longlin County were mainly comprised of woodland, ecological grassland, and agricultural production land. The area without rocky desertification was mainly concentrated in woodland and the proportion of the area without rocky desertification decreased in woodland and increased in grassland and agricultural production land. The area of potential rocky desertification was increasing in grassland and decreasing in woodland and agricultural production land area. The area of mild rocky desertification showed a decreasing trend in the proportion of woodland and an increasing trend in the proportion of grassland and agricultural production land area. The proportion of moderate rocky desertification area in woodland first increased and then decreased, and the proportion in grassland and agricultural production land continued to increase. The area of severe rocky desertification in woodland showed a tendency to decrease; in grassland, it showed a tendency to increase and then decrease; and in agricultural production land, it showed a tendency to increase.

In the land use transition, the land transfer area was basically in a rocky desertification area, among which moderate rocky desertification and severe rocky desertification accounted for more. The area of rocky desertification was increasing in the area of agricultural production land. Most of the areas where other land transformed into agricultural production land were moderate rocky desertification and severe rocky desertification areas. The rocky desertification in agricultural production land is due to the fact that humans destroy the soil fertility, surface vegetation, and erosion resistance, which accelerates soil erosion, thus indirectly driving the rocky desertification in the karst area [68]. The influence of woodland on the spatial distribution of rocky desertification is mainly the transition of woodland to industrial and mining production land, agricultural production land, and ecological grassland. Due to the reduction of vegetation cover, the landscape type gradually develops into monoculture and the stability of the karst ecosystem becomes less stable. The influence of grassland on the spatial distribution of rocky desertification is the transition of grassland to mining production land, agricultural production land, and ecological woodland. Grassland has a stronger ability to store water and retain soil. When grasslands are transformed into agricultural land, the vegetation on the ground is destroyed, and the intensity of soil erosion and land degradation are intensified, which increases the risk of rocky desertification [69]. When grasslands are converted into woodland, production land is transferred to ecological land, vegetation cover and biomass are increased, and the stability of ecosystems are enhanced [70]. In 2020, the eco-environment of rocky desertification improved and most of the production land that was converted into ecological land occurred in the area of no rocky desertification and potential rocky desertification.

### 4.3. Ecological Response to Land Use Change in Production-Living-Ecological Space

According to Equations (3) and (4), the EV data samples of karst mountains in 2005, 2010, 2015, and 2020 were routinely counted and spatially interpolated by using Arcgis 10.2. The regional eco-environmental quality of Longlin County was classified into lowest quality areas (EV < 0.5), lower quality areas (0.5 ≤ EV < 0.6), general quality areas (0.6 ≤ EV < 0.7), higher quality areas (0.7 ≤ EV < 0.75), and highest quality areas (EV ≥ 0.75).

#### 4.3.1. Spatial and Temporal Evolution of the Comprehensive Quality of Eco-Environment

(1) Time series change. The eco-environmental quality indices of Longlin County were 0.7, 0.699, 0.698, and 0.697, and the overall eco-environmental quality continued to decline during the study period. The differences in eco-environmental quality levels in the study area are obvious and most of them were in the general quality area and higher quality area, both of which accounted for more than 70% of the total area, indicating that the eco-environmental quality in the study area is acceptable. The highest quality area first increased and then decreased, with a trend of continuous decrease, but overall, the highest quality area increased. The higher quality area first decreased and then increased, and overall, the higher quality area decreased. The general quality area continued to decrease and then increased, and overall, the general quality area decreased. The lower quality area continued to increase, and the lowest quality area first continued to increase and then decreased, but overall, the area increased. In 2020, the lowest quality area increased 100% compared to 2005, and the lowest quality area increased rapidly. The lowest quality area, higher quality area, and highest quality area all changed more drastically in the early stage and changed more flatly in the later stage.

As shown in Table 4, the area of lowest quality ecological area in Longlin County doubled in 2020, which was mainly transferred from the lower quality area and general quality area. The lower quality area was mainly transferred from the general quality area. The highest quality area increased, mainly transferred from the higher quality area. The decrease in the general quality area was mainly transferred to the lower quality area and lowest quality area, and the higher quality area was transferred to the highest quality area. The polarization of the transfer of eco-environmental quality area also indicates that the low-level area of eco-environmental quality was on the rise and, significantly, explains the deterioration of the overall quality level of the region.

(2) Spatial divergence. The overall spatial pattern of the eco-environmental quality index for production-living-ecological space land use in Longlin County shows a spatial pattern characterized by an intermingling of the areas of highest quality, higher quality, and general quality, and by the general quality area surrounding the lower and lowest quality areas (Figure 2). The highest quality area was mainly located in Jinzhongshan Township in the southwest of Longlin County, where most of the Jinzhongshan Mountain Range is located, as well as in the mountainous area in the northeast and the mountainous area between Jieting Township and Yancha Township in the southwest. The functional land in the highest quality area was woodland, which was influenced by the topography and landscape, limiting the development of arable land, as well as industrial and mining development and urban agglomeration, and making the eco-environment of the highest quality. The general quality area was located in the town of De’e in the central-west of Longlin County. This town has a high elevation, numerous mountains and a fragmented landscape. The woodland and agricultural production land were intertwined. In a larger area in the north of the county, woodland, grassland, and agricultural production land were interspersed with each other and all kinds of land were easily found to be transferred to each other, making the eco-environment of an average quality. The lowest quality area was mainly the center of the county city, where urban living land, rural living land, and agricultural production land were interspersed. Due to the development of urbanization, the type of land use continued to shift and the scope of the lowest quality area continued to expand, which is basically consistent with the economic development of the county city and the scope of industrial expansion.

#### 4.3.2. The Affecting Ecological Effects Factors of Rock Desertification

By using Arcgis 10.2 to spatially overlay the eco-environmental quality index distribution data with the rocky desertification level distribution data, the eco-environment lowest quality and lower quality areas were mainly distributed in the rocky desertification area, and the mild rocky desertification area and moderate rocky desertification area both accounted for a relatively large proportion. The general quality eco-environment area was mainly distributed in the potential rocky desertification area, the mild rocky desertification area, and the moderate rocky desertification area. The highest quality area and higher quality area were mainly located in the area without rocky desertification, the potential rocky desertification area, and the mild rocky desertification area. In 2005, about 90% of the lowest quality area and lower quality area was in the rocky desertification area, and the area with moderate rocky desertification or above accounted for 50%. The area without rocky desertification, potential rocky desertification, and mild rocky desertification accounted for 71% of the highest quality eco-environment area and 58% of the higher quality area. By 2020, the rocky desertification area accounted for 75% and 76% of the lowest quality area and lower quality area, respectively. Additionally, the area below the mild rocky desertification level accounted for 91% and 85% of the highest quality and higher quality eco-environment areas, respectively, of which the area without rocky desertification accounted for a relatively large proportion. The results show that the level of rock desertification was lower in the higher quality eco-environment area as well as the highest quality area, the level of rock desertification was more serious in the lower quality eco-environment area and the lowest quality area, and the rock desertification areas were more widely distributed. Rocky desertification occurs in areas often accompanied by unreasonable land use structure and unreasonable transition of land use structure, causing natural disasters, such as soil erosion, that lead to the bareness of soil layer rocks, which in turn leads to the deterioration of the eco-environment.

#### 4.3.3. The Affecting Ecological Effects Factors of Land Use Transition

The eco-environmental impacts caused by land use transition are divided into positive and negative effects. According to equation 5, the changes of quality indices and contribution rates of the main functional land types leading to ecological improvement and degradation in Longlin County during 2005–2020 are given in Table 5. From 2005 to 2010, agricultural production land was transformed into woodland, watershed areas, and ecological grassland, while grassland was transformed into woodland and the increase of ecological land was the leading factor of ecological improvement. The functional land types that led to the improvement of eco-environment were relatively concentrated. The improvement of ecological quality caused by the transition of the first six land functions shown in the table accounts for 99.64%. On the contrary, the occupation of woodland and grassland by agricultural production land, urban and rural living land, industrial and mining production, land and ecological water areas, as well as the occupation of grassland to woodland, are important factors that led to the deterioration of the eco-environmental quality. The total contribution rate of these lands is 98.49%. The role of land function transition in the eco-environment shows the same trend in 2010–2015 and 2015–2020. During 2015–2020, industrial and mining production land was transformed into woodland to improve the quality of eco-environment. In summary, there are two trends of ecological improvement and deterioration that occurred at the same time in Longlin County, the outflow of woodland ecological land is the most important factor of eco-environment deterioration, the trend of eco-environment improvement is smaller than the trend of environmental deterioration, and the degree of eco-environment deterioration increased.

## 5. Discussion

### 5.1. Mechanisms of Land Use Transition and Eco-Environmental Effects in Karst Mountain Areas

Based on the previous studies, this work explores the changes in production-living-ecological space patterns in karst mountain counties, the incidence of rocky desertification in production-living-ecological space patterns, and the eco-environmental effects of production-living-ecological space. From 2005 to 2020, the spatial pattern of production-living-ecological space in the study area was relatively fragmented, and the land for living and industrial and mining production was constantly encroaching on the ecological land. In 2005, the area of industrial and mining production land and urban living land in the study area was relatively small, urbanization was not obvious, and it had been in a state of deep poverty. In 2002, Longlin County responded to the national western development strategy and fully implemented the return of cultivated land to forest. In the early stages of returning farmland to forest, the economic development of Longlin County, which had been mainly based on agriculture, was more difficult and the land use transition was more drastic in view of the reduction of land for agricultural production. Until 2010, rocky desertification in the study area had intensified, the eco-environment and economic development were poor, urban development was slow, and the urbanization rate was low. During the period of 2010–2020, the ecological development of the study area was combined with economic development, the land for living and industrial and mining production was increasing, the unreasonable land use development method gradually reduced, the retreat of cultivation to forest began to bear fruit, and the area of rocky desertification was decreasing. Most of the fallow land was sloping land with serious rocky desertification, sanding, soil erosion, and low and unstable food production. The study shows that, for karst mountain areas, strict implementation of forest resource management and protection, integrated soil erosion management, and rocky desertification management projects can effectively curb the serious deterioration of eco-environment in the process of competition between ecology and production and living land. In territorial spatial planning, it is still worth thinking about how the case study area can achieve a positive interaction between promoting economic and social development and ecological stability with a more rational and efficient land development and management model. We analyzed the transition of functional land use and the evolution of the spatial pattern of the national territory, which has a certain guiding significance for the delineation of the “three zones and three lines”. However, it is still lacking in examining the interaction between spatial evolution and social effects. Especially for the case area in this paper, after achieving poverty eradication, ways to prevent poverty return and guide the positive interaction between sustainable livelihoods of local residents and the optimization of territorial space allocation needs further in-depth study. Among those methods, the coordination of the “total control” and “moderate flexibility” of territorial spatial development is an important means to regulate the development of resources in karst mountain areas. It is particularly necessary to improve land use efficiency and explore more adaptive spatial layouts to reduce the continuous deterioration of rocky desertification and damage to living environments. These contents will be further developed in subsequent studies.

### 5.2. Strategies for Regulating Land Use Transition and Rocky Desertification in Karst Mountain Areas

In general, the ecological quality of this county is favorable; it has an acceptable ecological foundation, the highest ecological functions, such as climate regulation and soil conservation, and harmonious ecological and human socioeconomic development, which is of great significance to the regional ecological quality. However, it is also accompanied by the real problem of low residential carrying and non-agricultural production indicators in ecologically high-quality areas, thus resulting in a high number of poor people in the county. It is especially necessary to further explore strategies for high-ecological-quality regions to develop ecological economy and improve human living environments while focusing on maintaining ecological quality. People in karst mountain areas mostly live on mountains, and crops and cash crops are the main economic source for local residents. When improving the phenomenon of rocky desertification, the implementation of paid ecological fallowing, the teaching of economic forest planting techniques, and the protection of economic rights and interests of farmers who have lost their land are the key links in the coordinated development of ecology, production, and life. In the areas with average or low ecological quality, most of them are moderate and severe rocky desertification areas, which are also the areas with more active land use transition. In particular, the ecological land is transformed into production land and living land areas, and the expansion of production land and living land is the main reason for the deterioration of ecological quality. However, most of the karst mountain areas are not mature enough for urbanization; there are still contradictions between economic growth, land expansion, and ecological protection; the ecological space can still be restricted by the production as well as the living space; and the environmental quality is prone to continuous deterioration. Developing mountainous characteristic high-efficiency agriculture and green ecological agriculture, accounting for the improvement of rocky desertification, exploring a new path for the development of characteristic ecological agriculture in rocky desertification areas, expanding the income sources of regional farmers, and preventing the return to poverty are still the basic problems faced by this case study area.

The remote sensing image data used in this study have time differences, and although it is considered to have less of an impact on the accuracy of the research results after checking with related information, objectively, there are still some errors. In addition, this work takes a natural ecological perspective and does not delve into the analysis of the driving forces of land use transition under social factors. We will collect further data and expert opinions to deepen the research-related contents.

## 6. Conclusions

Using econometric statistics, remote sensing, and GIS technology combined with relevant research at home and abroad, this study comprises a spatial-temporal analysis on the land use transition, rocky desertification evolution, and eco-environmental effects of production-living-ecological space in Longlin County, in karst mountain areas from 2005 to 2020.

The results show that, from 2005 to 2020, the land use transition of production-living-ecological space in Longlin County is mainly manifested in the decrease of ecological land area and production land area, and in the rapid increase of living land area. According to the secondary land use categories, the area of agricultural production land decreased, the area of water ecological land greatly increased, the area of woodland and grassland continued to decrease, and the area of urban and rural living land greatly increased.

From 2005 to 2020, the land desertification in Longlin County is mainly characterized by potential, mild, and moderate rocky desertification. In general, the area without rocky desertification increased, the area with rocky desertification decreased, the eco-environment of rocky desertification improved, and the treatment of rocky desertification was quite effective. Among the land use types of production-living-ecological space, the proportion of agricultural production land in areas with mild rocky desertification and severe rocky desertification is gradually increasing, and the proportion of woodland in areas without rocky desertification and potential rocky desertification is gradually increasing. In the process of converting woodland and grassland into agricultural production land, soil fertility decreases and surface vegetation decreases, which accelerates the development of rocky desertification.The ecological quality index of Longlin County decreased from 0.7 in 2005 to 0.697 in 2020, with a slight deterioration in overall quality. From 2005 to 2020, the area of the higher quality zone accounted for about 40%, constituting the main body of production-living-ecological space, and the overall eco-environment of Longlin County was of good quality. The area and proportion of the high-quality area and low-quality area continued to increase, showing a trend of polarization and continuous expansion. Among them, the rapid expansion of urban and rural living land as well as industrial and mining production land is the main factor for the continuous expansion of low-quality areas.The low-quality eco-environment areas of in Longlin county are mainly distributed in rocky desertification areas, and the most important areas are severe rocky desertification areas and moderate rocky desertification areas. High-quality areas are mainly distributed in areas without rocky desertification and with potential rocky desertification. The unreasonable land use structure and the unreasonable land use transition led to the development of rocky desertification, which is one of the factors that led to the deterioration of the eco-environment.From 2005 to 2020, there were two trends of ecological improvement and deterioration in Longlin County, a karst mountain area. The trend of eco-environment improvement was less than that of environmental deterioration, and the degree of eco-environment deterioration was increasing. The transition of agricultural production land into ecological land, such as woodland, water, and grassland, as well as the transition of grassland into woodland, is the leading factor in the improvement of eco-environment. The occupation of woodland and grassland by agricultural and industrial and mining production land, urban and rural living land, water ecological land, and grassland to woodland transition areas are important factors leading to the deterioration of eco-environmental quality.

In summary, the current exploration of land use change and ecological effects adopts a combination of qualitative and quantitative approaches, focusing on the degree of carrying capacity, vulnerability of geographic environmental elements and their spatial combinations. On this basis, a large number of studies on the interaction between the suitability of territorial space development and environmental carrying capacity on territorial space have proven to be feasible. However, the comprehensive study of land use transition, ecological effects, and rocky desertification management in the production-living-ecological space of karst mountain areas still needs to be explored in depth, especially for the spatial regulation strategies of karst mountain areas under different rocky desertification levels, which is urgent to supplement. This paper quantitatively identifies the evolution and eco-environmental effects of rocky desertification under the land use transition of production-living-ecological space. The research methods and results are helpful to promote the rational allocation and fine management of land resources in karst mountain areas, and provide basic data and some theoretical references for the optimization of territorial space and the delineation of the production-living-ecological space in karst mountain areas. Meanwhile, the policies on land development and rocky desertification management since 1992 were reviewed (Figure 5), and the real effects of their implementation were summarized. Although the rocky desertification management strategy in the case area has some lag, the ecological management is quite effective and provides reference for other karst areas. In urbanization, the problems that mountain areas and woodland reduced, such as insufficient investment in rocky desertification, policies’ lack of resident’s needs, and waste of rural quality land resources due to ecological migration [71], should be given more attention in rocky desertification management, ecological restoration, and territorial spatial planning.

## Figures and Tables

**Figure 1 ijerph-19-07587-f001:**
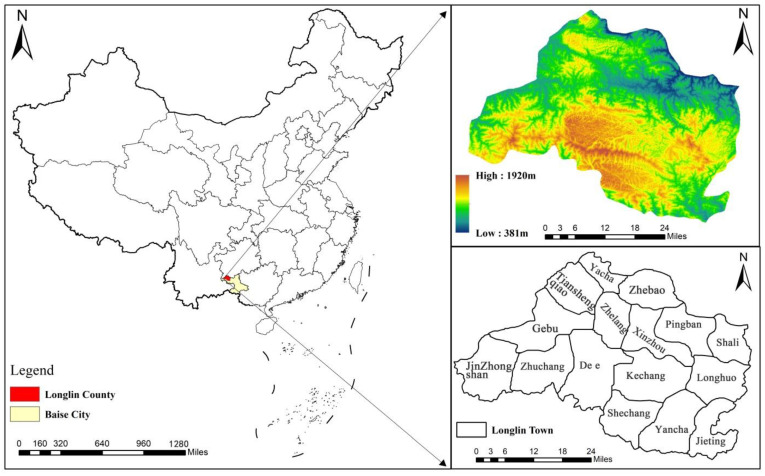
Location, topography, and administrative division of the study area.

**Figure 2 ijerph-19-07587-f002:**
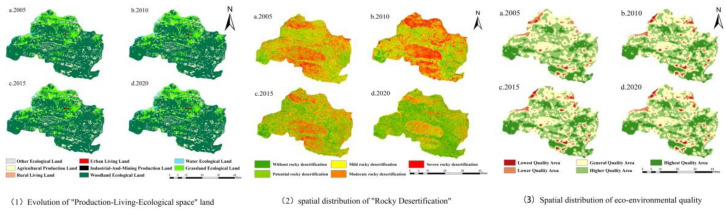
Evolution of production-living-ecological space land use, rocky desertification, and eco-environmental quality in Longlin County from 2005 to 2020.

**Figure 3 ijerph-19-07587-f003:**
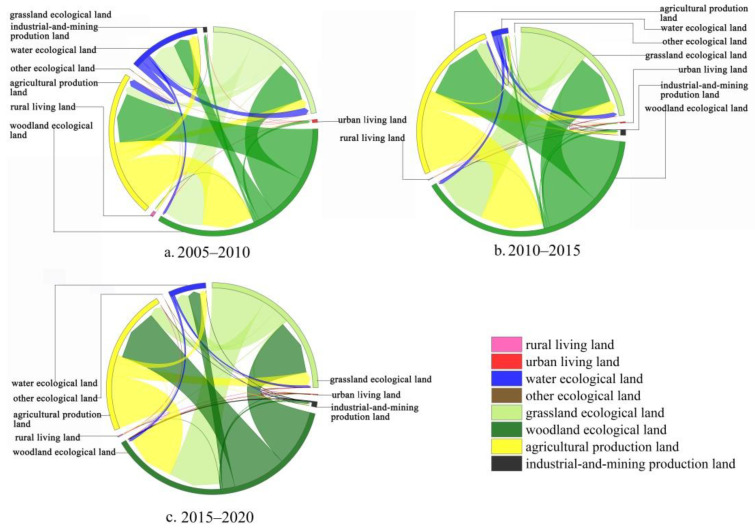
Land transfer matrix of production-living-ecological space from 2005 to 2020.

**Figure 4 ijerph-19-07587-f004:**
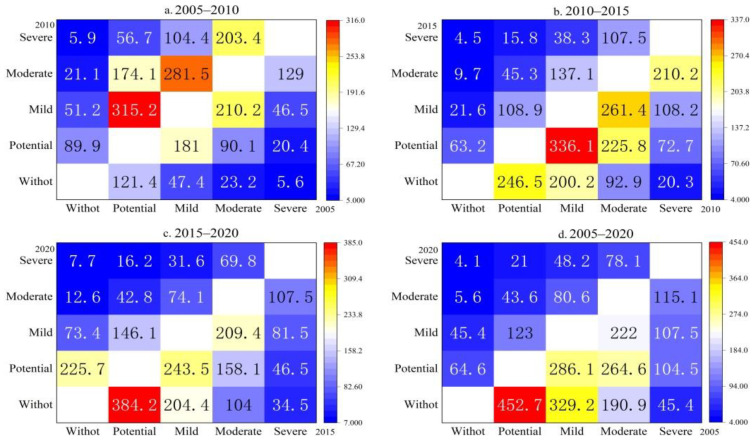
Transfer matrix of rocky desertification area in Longlin from 2005 to 2020 (km^2^).

**Figure 5 ijerph-19-07587-f005:**
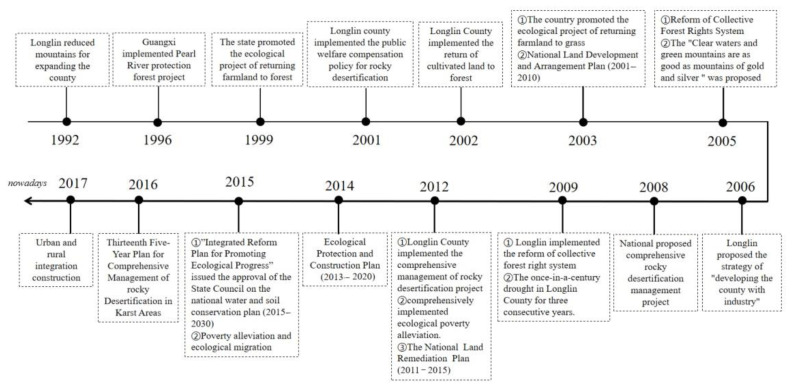
Policy factors affecting land change and rocky desertification in Longlin County in a recent 20-year period.

**Table 1 ijerph-19-07587-t001:** Land use classification of production-living-ecological space and eco-environmental quality index in Longlin County.

Primary Land Use Types	Secondary Land Use Types	Secondary Land Use Types	Eco-Environmental Quality Index Assignment
Production Land	Agricultural Production Land	Paddy field, dry land	0.26
	Industrial and mining Production Land	Other construction land	0.15
Ecological Land	Woodland Ecological Land	Wooded land, shrubland, open woodland, other woodland	0.77
	Grassland Ecological Land	High-cover grassland, medium-cover grassland, low-cover grassland	0.69
	Water Ecological Land	Rivers and canals, lakes, reservoirs and ponds, beaches and mudflats	0.55
	Other Ecological Land	Bare land, bare rocky land	0.015
Living Land	Urban Living Land	Urban land	0.2
	Rural Living Land	Rural settlements	0.2

**Table 2 ijerph-19-07587-t002:** Land area of production-living-ecological space in Longlin County and its changes from 2005 to 2020 (km^2^).

Year	Grassland Ecological Land	Urban Living Land	Industrial and Mining Productionland	Woodland Ecological Land	Rural Living Land	Agricultural Production Land	Other Ecological Land	Water Ecological Land
2005	533.8	1.7	0	2583.6	0.1	386.9	0.06	31.8
2010	525.4	2.8	1.5	2575.5	1.2	385	0	46.6
2015	525.1	3.1	3.7	2576.3	1.2	382.6	0.1	45.8
2020	521.6	4.2	5.8	2566.9	2.4	378.9	0	58.1
2005–2010	−8.4	1.1	1.5	−8.1	1.1	−1.9	−0.06	14.8
Range of change	−0.315%	12.941%		−0.063%	220.000%	−0.098%	−20.000%	9.308%
2010–2015	−0.3	0.3	2.2	0.8	0	−2.4	0.1	−0.8
Range of change	−0.011%	2.143%	29.333%	0.006%	0.000%	−0.125%		−0.343%
2015–2020	−3.5	1.1	2.1	−9.4	1.2	−3.7	−0.1	12.3
Range of change	−0.133%	7.097%	11.351%	−0.073%	20.000%	−0.193%	−20.000%	5.371%
2005–2020	−12.2	2.5	5.8	−16.7	2.3	−8	−0.06	26.3
Range of change	−0.152%	9.804%		−0.043%	153.333%	−0.138%	−6.667%	5.514%

**Table 3 ijerph-19-07587-t003:** Areas of different rocky desertification levels and changes from 2005 to 2020 in Longlin County (km^2^).

Rocky Desertification Level	Year	Area (km^2^)	Land Use Types				2005–2010		2010–2015		2015–2020	
			Grassland Ecological Land (%)	Industrial and Mining Production Land (%)	Woodland Ecological Land (%)	Agricultural Production Land (%)	Area of Change	Dynamic Attitude (%)	Area of Change	Dynamic Attitude (%)	Area of Change	Dynamic Attitude (%)
**No rocky desertification**	**2005**	258	9.72%	0.00%	81.52%	5.62%	29.86	2.31%	460.74	32.01%	407.78	10.89%
	**2010**	287.86	6.95%	0.00%	84.16%	4.05%						
	**2015**	748.59	9.86%	0.00%	84.27%	5.84%						
	**2020**	1156.38	11.71%	0.03%	80.11%	7.90%						
**Potential rocky** **desertification**	**2005**	949.67	13.20%	0.00%	75.51%	10.82%	−285.72	−6.02%	280.74	8.46%	84.49	1.79%
	**2010**	663.95	13.18%	0.01%	75.59%	10.49%						
	**2015**	944.69	12.98%	0.02%	75.83%	8.27%						
	**2020**	1029.18	14.42%	0.06%	74.78%	9.67%						
**Mild rocky desertification**	**2005**	952.08	17.73%	0.00%	68.22%	13.48%	9.33	0.20%	−212.19	−4.41%	−43.22	−1.15%
	**2010**	961.41	16.54%	0.11%	67.99%	14.56%						
	**2015**	749.22	16.92%	0.22%	66.76%	15.29%						
	**2020**	706.00	17.67%	0.49%	60.20%	15.96%						
**Moderate rocky desertification**	**2005**	889.73	17.48%	0.00%	69.82%	12.53%	79.13	1.78%	−285.58	−5.90%	−304.42	−8.91%
	**2010**	968.85	17.30%	0.03%	70.15%	12.33%						
	**2015**	683.28	18.82%	0.10%	67.18%	13.57%						
	**2020**	378.86	20.02%	0.30%	64.61%	13.70%						
**severe rocky desertification**	**2005**	442.37	20.26%	0.00%	67.44%	12.19%	169.06	7.64%	−245.38	−8.03%	−144.64	−7.90%
	**2010**	611.43	22.09%	0.03%	66.71%	10.92%						
	**2015**	366.05	21.94%	0.12%	64.58%	12.93%						
	**2020**	221.41	17.17%	0.79%	64.73%	16.50%						

**Table 4 ijerph-19-07587-t004:** Area transfer matrix of eco-environmental quality zoning in Longlin County from 2005 to 2020 (km^2^).

	2005				
2020	Lowest Quality Area	Lower Quality Area	General Quality Area	Higher Quality Area	Highest Quality Area
lowest quality area	5.5	17.5	14.75	1.5	0.25
lower quality area	7.25	93.75	102.5	19.25	0
general quality area	5.25	67	852.75	282	15
higher quality area	1.75	25.5	245.5	1027.75	105.25
highest quality area	0	0	20.75	161.75	462.75

**Table 5 ijerph-19-07587-t005:** Major land use transition affecting eco-environmental quality and their contribution rates from 2005 to 2020.

Patterns	Land Use Function Transition	Contribution Rate	Contribution Percentage
Lead to eco-environment improvement	**2005–2010**		
	Agricultural production land to Woodland Ecological Land	0.00309	69.57%
	Agricultural production land to Grassland Ecological Land	0.00046	10.45%
	Grassland Ecological Land to Woodland Ecological Land	0.00038	8.52%
	Agricultural production land to Water Ecological Land	0.00032	7.24%
	Water Ecological Land to Grassland Ecological Land	0.00011	2.54%
	Water Ecological Land to Woodland Ecological Land	0.00006	1.31%
	Total		99.64%
	**2010–2015**		
	Agricultural production land to Woodland Ecological Land	0.00422	74.14%
	Agricultural production land to Grassland Ecological Land	0.00066	11.66%
	Grassland Ecological Land to Woodland Ecological Land	0.00054	9.52%
	Water Ecological Land to Woodland Ecological Land	0.00009	1.61%
	Water Ecological Land to Grassland Ecological Land	0.00007	1.28%
	Agricultural production land to Water Ecological Land	0.00007	1.15%
	Total		99.36%
	**2015–2020**		
	Agricultural production land to Woodland Ecological Land	0.00621	66.15%
	Agricultural production land to Grassland Ecological Land	0.00135	14.43%
	Grassland Ecological Land to Woodland Ecological Land	0.00098	10.47%
	Agricultural production land to Water Ecological Land	0.00042	4.49%
	Industrial and mining Production Land to Woodland Ecological Land	0.00016	1.67%
	Water Ecological Land to Woodland Ecological Land	0.00011	1.17%
	Total		98.38%
	**2005–2010**		
Lead to eco-environment degradation	Woodland Ecological Land to Agricultural production land	−0.00304	55.27%
	Grassland Ecological Land to Agricultural production land	−0.00047	8.52%
	Grassland Ecological Land to Water Ecological Land	−0.00043	7.80%
	Woodland Ecological Land to Water Ecological Land	−0.00042	7.63%
	Woodland Ecological Land to Grassland Ecological Land	−0.00039	7.00%
	Water Ecological Land to Agricultural production land	−0.00024	4.42%
	Woodland Ecological Land to Urban Living Land	−0.00018	3.36%
	Woodland Ecological Land to Industrial and mining Production Land	−0.00018	3.35%
	Grassland Ecological Land to Industrial and mining Production Land	−0.00006	1.12%
	Total		98.49%
	**2010–2015**		
	Woodland Ecological Land-Agricultural production land	−0.00406	70.40%
	Grassland Ecological Land to Agricultural production land	−0.00067	11.53%
	Woodland Ecological Land to Grassland Ecological Land	−0.00053	9.11%
	Woodland Ecological Land to Industrial and mining Production Land	−0.00021	3.56%
	Woodland Ecological Land to Water Ecological Land	−0.00007	1.25%
	Woodland Ecological Land to Urban Living Land	−0.00006	1.04%
	Total		96.88%
	**2015–2020**		
	Woodland Ecological Land-Agricultural production land	−0.00624	60.83%
	Grassland Ecological Land-Agricultural production land	−0.00138	13.46%
	Woodland Ecological Land to Grassland Ecological Land	−0.00115	11.18%
	Grassland Ecological Land to Water Ecological Land	−0.00042	4.11%
	Woodland Ecological Land to Industrial and mining Production Land	−0.00029	2.83%
	Woodland Ecological Land to Urban Living Land	−0.00021	2.02%
	Grassland Ecological Land to Industrial and mining Production Land	−0.00019	1.87%
	Woodland Ecological Land to Rural Living Land	−0.00015	1.47%
	Total		97.77%

## Data Availability

The data used to support the findings of this study are available from the corresponding author upon request.

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
