# Peer review of "Land Use Transition and Eco-Environmental Effects in Karst Mountain Area Based on Production-Living-Ecological Space: A Case Study of Longlin Multinational Autonomous County, Southwest China"

_ijerph, 2022, doi:10.3390/ijerph19137587_

Round 1

Reviewer 1 Report

Manuscript: Land use transition and eco-environmental effects in karst mountain area based on Production-Living-Ecological space: A case study of Longlin Multinational Autonomous County, Southwest China

The manuscript is a case study of land use transition and eco-environmental effects based on Production-Living-Ecological in a specific geo-environment (karst mountain) from 2005 to 2020.   

The study addresses an important research topic in sustainability: land use and the environmental effects. The vast period covered by the study, the large number of data, the methods used (remote sensing and GIS technology) give to the research scientific soundness for publication. However, the manuscript requires some degree of revision.

Authors may consider the following comments:

As rocky desertification is the predominant process in the area, information about geology including a lithological characterization would fit well in the description of the area. The study presents vast amounts of geospatial data, but field data are missing.  Due to the large area studied (3551km2) field data collection is not possible, but authors could include citations on regional and local geological-geomorphological studies. Same comment for geologic hazards, mainly landslides.

A general revision throughout the manuscript is needed to eliminate typos (examples: lines 19; 102; 204; 761) and repetitions (example: lines 233-238).

The paper would benefit of English editing (examples: lines 14-20- punctuation; lines 224-225, data are).

Author Response

Dear editor:

Thank you for your latter and advice. We are very pleased to be asked to submit a revision. Those comments are all valuable and very helpful for revising and improving our paper, as well as the important guiding significance to our researches.We have revised the paper point by point.The revision is marked in red.We hope that the correction will meet with approval.

Thanks again for your reconsideration of our manuscript. We look forward to your favorable decision.

Kind regards,

Kongtao Qin

 Min Wang

Point 1:As rocky desertification is the predominant process in the area, information about geology including a lithological characterization would fit well in the description of the area.The study presents vast amounts of geospatial data, but field data are missing.  Due to the large area studied (3551km2) field data collection is not possible, but authors could include citations on regional and local geological-geomorphological studies.Same comment for geologic hazards, mainly landslides.

Response 1:

For your suggestions, we have modified the chapter 3.1 study area(line200-241). We have supplemented the geological, topographical and geomorphic information of the study area, as well as its climate information, and also described the geological disasters and causes that are easy to form. And some geological disasters affected by rocky desertification are described in the last paragraph of the introduction.

Point 2: A general revision throughout the manuscript is needed to eliminate typos (examples: lines 19; 102; 204; 761) and repetitions (example: lines 233-238).

Response 2: 

Sorry for the inconvenience of reading this due to my carelessness, I have corrected the spelling errors in the entire text.

Point 3: The paper would benefit of English editing (examples: lines 14-20- punctuation; lines 224-225, data are).

Response 3:

Thank you for your advice, I have corrected these grammatical errors and punctuation errors.

In addition, the abstract of this paper has been completely modified, and the introduction has been greatly modified, which expanded the discussion scope of the literature review.At the end of the article, we summarized the policy factors and major natural disasters that affected land use and rocky desertification control in the study area. And also discussed the effects and deficiencies of policy implementation, as well as the significance and deficiencies of this paper.Please see the attachment.

Reviewer 2 Report

Good paper, almost ready for publication. I have only minor comments/suggestions to offer below.

Lines 78-80 - what about various marine and land resources in this context? They also can and will be affected and impacted.

Last paragraph in section 1: Some references will be quite appropriate at the end of lines 102-104 and line 107.

Section 3.1. Study area. Line 204 - please correct the typo, should be "Area".

Also, couple of sentences on general karst geology and properties in the study area seem to be quite appropriate in this section. This information can be very helpful to a general reader.

Figures 2 and 6 - legend boxes and accompanying text within the figures is barely readable. Could you please enlarge?

Author Response

Dear Reviewer:

Thank you for your latter and advice. We are very pleased to be asked to submit a revision. Those comments are all valuable and very helpful for revising and improving our paper, as well as the important guiding significance to our researches.We have revised the paper point by point.The revision is marked in red.We hope that the correction will meet with approval.

Thanks again for your reconsideration of our manuscript. We look forward to your favorable decision.

Kind regards,

Kongtao Qin

 Min Wang

Point 1:Lines 78-80 - what about various marine and land resources in this context? They also can and will be affected and impacted.

Response 1:

According to your suggestions, we have made a lot of modifications in the introduction to expand the scope of literature review. In the second paragraph of the introduction(line63-71, reference1-17), we have made a series of summaries of the influencing factors brought about by land use change.

Point 2: Last paragraph in section 1: Some references will be quite appropriate at the end of lines 102-104 and line 107.

Response 2: 

According to your suggestions, this part is in the third paragraph of the introduction. In this part(line95-106,reference30-39), we have supplemented the impact of rocky desertification and added references.

Point 3: Section 3.1. Study area. Line 204 - please correct the typo, should be "Area".

Response 3:

Sorry for the inconvenience of reading this due to my carelessness, I have corrected the spelling errors in the entire text.

Point 4: Also, couple of sentences on general karst geology and properties in the study area seem to be quite appropriate in this section. This information can be very helpful to a general reader.

Response 4:

For your suggestions, we have modified the chapter 3.1 study area(line200-241). We have supplemented the geological, topographical and geomorphic information of the study area, as well as its climate information, and also described the geological disasters and causes that are easy to form. And some geological disasters affected by rocky desertification are described in the last paragraph of the introduction.

Point 5: Figures 2 and 6 - legend boxes and accompanying text within the figures is barely readable. Could you please enlarge?

Response 5: Thank you for your suggestion. I have enlarged the font of the picture

In addition, the abstract of this paper has been completely modified, and the introduction has been greatly modified, which expanded the discussion scope of the literature review.At the end of the article, we summarized the policy factors and major natural disasters that affected land use and rocky desertification control in the study area. And also discussed the effects and deficiencies of policy implementation, as well as the significance and deficiencies of this paper.

Reviewer 3 Report

The presents the results of a study on the land use transition with the special focus on the territory of China.

The literature and theoretical background reflect almost exclusively Chinese sources. To give the study a more meaning for the international research community it is advisable to broaden the scope of literature review and discussion.

In the chapter 2.2 it is not clear which theory the authors are addressing and why. Is it really necessary to explain it this way? In the current form it is just a general explanation of the definition od sustainability. Could be shorter.

110-116 – the sentence is very long but not complete and not understandable.

Please provide a clear statement on the research questions or purpose of the study. It is somehow there but it could be helpful to put in more straight forward and clear.

Please shorten and rewrite the abstract, it goes to much into the details, the main overview message gets lost.

Line 183: consier if the sentence really makes sense: “mainly positive and negative”?

Figure 1: it is great, that the authors provide an introduction to the location of the case study area!

Figure 3: it is a very interesting graphical representation of the transitions in the land use. It would be maybe easier to understand if the land use types were written by the circle sections? It is just a suggestion, not a demand for a change, I live it up to you.

Figure 4: in this figure, some more explanation is needed – the categories (and what they relate to) should be explained also in the picture.

The discussion in the current form is mainly a summary of the results. It is good to provide this summary, but a scientific discussion putting the outcomes of the study into the context of other results and theories is missing. Did other studies (or hypothesis) are similar or different to this one? What do we learn from the study in the meaning of what is new to the international research community?

Author Response

Dear Reviewer:

Thank you for your latter and advice. We are very pleased to be asked to submit a revision. Those comments are all valuable and very helpful for revising and improving our paper, as well as the important guiding significance to our researches.We have revised the paper point by point.The revision is marked in red.We hope that the correction will meet with approval.

Thanks again for your reconsideration of our manuscript. We look forward to your favorable decision.

Kind regards,

Kongtao Qin

 Min Wang

Point 1:The literature and theoretical background reflect almost exclusively Chinese sources. To give the study a more meaning for the international research community it is advisable to broaden the scope of literature review and discussion.

Response 1:

According to your suggestions, we have made a lot of modifications in the introduction to expand the scope of literature review. Combined with the international background literature, in the second paragraph of the introduction, we summarize the influencing factors of land use transition, the existing research on land use transition (line63-71, reference1-17). In the third paragraph of the introduction, we also have supplemented the impact of rocky desertification and added references(line95-106,reference30-39).

Point 2: In the chapter 2.2 it is not clear which theory the authors are addressing and why. Is it really necessary to explain it this way? In the current form it is just a general explanation of the definition od sustainability. Could be shorter.

Response 2: 

According to your suggestion, we have deleted the superfluous theoretical description in this chapter.

Point 3: 110-116 – the sentence is very long but not complete and not understandable。

Response 3:

Sorry for the inconvenience caused to your reading. I have revised this sentence and clearly pointed out the research purpose (line106-115). Specific modifications are as follows:

Therefore, this paper analyzes the territorial spatial changes of Longlin Multinational Autonomous County from 2005 to 2020.The research aims, 1) to determine the direction of land use transition by analyzing land type changes through remote sensing images, to clarify the action mechanism and competition process among the three functional space of "Production-Living-Ecological". And to assess the evolutionary direction of rocky desertification and the quality of eco-environment. 2) Accurately grasp the evolution , distribution characteristics of the three functional space and their driving mechanisms, which provide case support and countermeasure suggestions for territorial space optimizing , poverty alleviation and high-quality development.

Point 4: Please provide a clear statement on the research questions or purpose of the study. It is somehow there but it could be helpful to put in more straight forward and clear.

Response 4:

According to your suggestion, we rewrote the abstract to highlight the purpose and significance of our research. In the last paragraph of introduction (line 106-115), we directly describe the research purpose of the article,as the Response 3.

Point 5: Please shorten and rewrite the abstract, it goes to much into the details, the main overview message gets lost.

Response 5:

According to your suggestion, we rewrote the abstract to highlight the purpose and significance of our research.Specific modifications are as follows:

The linkage mechanisms and optimization strategies between land use transition and eco-environmental effects that occurring in the Production-Living-Ecological space of karst mountainous areas remain under-explored in the current literature. Based on county data collected in Longlin Multinational Autonomous County of Guangxi, which is located in the rocky desertification area of Yunnan, Guangxi and Guizhou, this study contributes a county-level analysis on land use transition and eco-environmental effects in karst mountain county by addressing two research questions: 1) Which factors of land use transition are related to the eco-environmental effects of Production-Living-Ecological space? 2) What are the key land allocation mechanisms behind the interventions of local rocky desertification regulation policies? Two sets of analyses were conducted to address these two questions, quantitative analysis of spatial and temporal evolution beteen land use transition, rocky desertification, and its eco-environmental effects, and qualitative analysis of policy interventions in Production-Living-Ecological land development and rocky desertification management. The findings show that the occurrence of rocky desertification accompanied by the unreasonable land use structure transition and ecological land squeezed by production-living land is an important factor. Specifically, coordinating ecological and socio-economic in urbanization strategy is significance to karst mountains areas. Moreover, the orderly increase of woodland slow down rocky desertification. Policies of "returning farmland to forest" and "afforestation of wasteland" have significantly reduced rocky desertification that effect the rocky management, and can be applied to other geographical situations.

Point 6: Line 183: consier if the sentence really makes sense: “mainly positive and negative”?

Response6:

We are sorry that this sentence is not clear. This sentence is a bit superfluous, so we have deleted it.

Point7: it is a very interesting graphical representation of the transitions in the land use. It would be maybe easier to understand if the land use types were written by the circle sections? It is just a suggestion, not a demand for a change, I live it up to you.

Response7:

Thank you for your suggestion. I have tried to describe the information on the map completely and marked the land use type on the map.

Point8: Figure 4: in this figure, some more explanation is needed – the categories (and what they relate to) should be explained also in the picture.

Response8:

Thank you for your suggestion. I have described the information on the map completely, and marked out the type of rocky desertification and the area proportion of each rocky desertification level area.

Point9: The discussion in the current form is mainly a summary of the results. It is good to provide this summary, but a scientific discussion putting the outcomes of the study into the context of other results and theories is missing. Did other studies (or hypothesis) are similar or different to this one? What do we learn from the study in the meaning of what is new to the international research community?

Response9:

 We added a discussion at the end of the conclusion of the article (line826-850), which is about the policy factors affecting land use and rocky desertification control, and the significance of this study. Specific modifications are as follows:

In summary, the current exploration of land use change and ecological effects adopts a combination of qualitative and quantitative approaches, focusing on the degree of carrying capacity, vulnerability of geographic environmental elements and their spatial combinations. On this basis, a large number of studies on the interaction between the suitability of territorial space development and environmental carrying capacity on territorial space have been proved to be feasible. However, the comprehensive study of land use transition, ecological effects and rocky desertification management in "Production-Living-Ecological space" of karst mountains areas still needs to be explored in depth; especially for the spatial regulation strategies of karst mountains areas under different rocky desertification levels, which is urgent to supplement. This paper quantitatively identifies the evolution and eco-environmental effects of rocky desertification under land use transition of "Production-Living-Ecological space". The research methods and results are helpful to promote the rational allocation and fine management of land resources in karst mountains areas, and provide basic data and some theoretical references for the optimization of territorial space and the delineation of the "Production-Living-Ecological space" in karst mountains areas. Meanwhile, the policies on land development and rocky desertification management since 1992 are reviewed (Figure 8), and the real effects of their implementation are summarized. Although the rocky desertification management strategy in the case area has some lag, the ecological management is quite effective and provide reference for other karst areas. In the urbanization, the problems that mountain and woodland reduced, insufficient investment in rocky desertification, policies lack of residents needs, waste of rural quality land resources due to ecological migration, should be given more attention in rocky desertification management, ecological restoration and territorial spatial planning .

This manuscript is a resubmission of an earlier submission. The following is a list of the peer review reports and author responses from that submission.